# Hygienic Quality of Air-Packed and Refrigerated or Frozen Stored Döner Kebab and Evaluation of the Growth of Intentionally Inoculated *Listeria monocytogenes*

**DOI:** 10.3390/microorganisms13040701

**Published:** 2025-03-21

**Authors:** Francesca Coppola, Giada Ferluga, Lucilla Iacumin, Cristian Bernardi, Michela Pellegrini, Giuseppe Comi

**Affiliations:** 1Food Sciences Institute, National Research Council, Via Roma, 64, 83100 Avellino, Italy; fracop93@gmail.com; 2Department of Agricultural, Food, Environmental and Animal Science, University of Udine, Via Sondrio 2/a, 33100 Udine, Italy; ferluga.giada@spes.uniud.it (G.F.); lucilla.iacumin@uniud.it (L.I.); 3Department of Veterinary Medicine and Animal Sciences, University of Milan, 20122 Lodi, Italy; cristian.bernardi@unimi.it

**Keywords:** döner kebab, hygienic quality, *Listeria monocytogenes*

## Abstract

Döner kebab, a meat product of Middle-Eastern origin, has gained significant popularity and is now widely consumed across Europe. The recipe varies depending on the area, with beef, turkey, lamb, or chicken being used as main ingredients. The aim of this work was to assess the hygienic-sanitary quality of raw and cooked döner kebabs stored at 4 ± 2 °C for 10 days and at 8 ± 2 °C for the next 20 days or frozen (−18 °C) for one month. One additional aim was to determine the potential growth of *Listeria monocytogenes* intentionally inoculated in cooked döner kebab during storage at 4 ± 2 °C or freezing. The concentration of Total Viable Count (TVC) and the Enterobacteriaceae of the 100 samples of raw döner kebab were less than 7 log CFU/g and 4 log CFU/g, respectively. Consequently, the samples can be considered acceptable and similar to traditional raw meat. The cooked döner kebab can be considered safe for a period of 30 days, especially from a microbiological point of view, when stored under refrigerated conditions, also taking into account possible thermal abuse. Coagulase Positive Cocci (CPC), *Clostridium* H_2_S^+^, *Salmonella* spp., and *Listeria monocytogenes* were never found in any of the samples. After 30 days, the TVC was at the level of 6 log CFU/g and Enterobacteriaceae at less than 4 log CFU/g. The main concern was related to microbial or tissue activity, resulting in an increase in total volatile basic nitrogen (TVB-N) content. However, in the cooked samples, the TVB-N content remained below 40 mg N/100 g at the end of the shelf-life period (32.5 mg N/100 g), which is still considered an acceptable value. In addition, the level of Malondialdehyde (MDA) was found to be within acceptable limits, with a reading of 1.4 nmol/g attained after 30 days. The same product, when frozen and stored at −18 °C, can be considered stable for a minimum of 6 months, both from a microbiological and a physico-chemical point of view. No microbial growth was observed. The TVB-N and the MDA levels increased, but after 6 months, their levels were still acceptable, with values of 19.1 mg N/100 g and 1.2 nmol/g, respectively. These observations demonstrate low protein degradation and lipid oxidation during the shelf-life period. The challenge test showed that *Listeria monocytogenes* did not grow in döner kebab either when stored at 4 ± 2 °C for 10 days and 8 ± 2 °C for 20 days or when stored at −18 °C for 6 months. The concentration of *L. monocytogenes* was found to be 5.4 log CFU/g in the refrigerated products and 4.9 log CFU/g in the frozen products. At the end of the shelf-life period, the *L. monocytogenes* load in both products was lower than the initial concentration that had been added. Finally, the use of air-packaging has been proven to be beneficial to the preservation of the product and maintained its microbiological and physico-chemical properties intact. Despite these good results, future directions could be to investigate different plastic films and packaging such as Modified Atmosphere (MAP), Vacuum (VP), and Sous Vide packaging (SVP).

## 1. Introduction

Döner kebab, a meat product of Middle-Eastern origin, has gained significant popularity and is now widely consumed throughout Europe. The name of döner is derived from the Turkish dönmek and refers to the cooking method, which involves rotating large cones made of meat and other ingredients in open ovens, allowing them to cook slowly on the surface [1]. Döner kebab born in the 1800s in Turkey and began to spread in Europe with the great wave of Turkish immigration, starting in the 1950s. In many European countries, including Italy, kebab outlets number in the tens of thousands today. In Germany, in 2020, 400 tonnes of kebab were produced per day, generating a turnover of EUR 4.5 billion per year [1]. Döner kebab is typically sold and consumed in small retail outlets, where operators are used to purchasing the cones—usually in a frozen state—from industrial plants and completing the process in open ovens. The sliced meat is commonly served in sandwiches, together with a variety of vegetables and sauces, providing a convenient and inexpensive meal.

The traditional production process entails slicing and/or mincing various cuts of meat, which are then marinated before being stacked to form a cone. There are a variety of recipes used for the marinating: spices such as oregano, mint, chilli, cinnamon, cumin, coriander, and pepper are used, as well as salt, oil, onions, onion juice, chopped tomatoes, lemon juice, milk or milk powder, yoghurt, egg, grape juice, and white sugar. Due to the moisture and fat content, meat fragments tend to adhere to each other, especially if the mass is frozen. During cooking, coagulation and other heat-induced processes increase the level of cohesion. The cooking method involves inserting a steel skewer in the centre of the cone, holding it vertically, and slowly rotating it on its axis in front of a grill. Normally, meat is initially frozen or refrigerated. The meat slices are gradually removed from the surface of the cone, thereby exposing less-cooked portions to the heat source in order to be cooked. Meat slices are characterized by a crunchy, more cooked exterior and a more tender and moister interior [2,3,4].

To produce döner kebab, different types of meat may be used: chicken, turkey, lamb, or beef. In Italy, retail outlets frequently purchase frozen raw material from national industrial plants, and at other times from local butchers. In addition to meat, ingredients may include stabilisers (E450 diphosphates, E451 triphosphates, E331 sodium citrate, E500 sodium carbonate), antioxidants (E300 ascorbic acid, E301 sodium ascorbate), flavour enhancers (E621 sodium glutamate), preservatives (E250 sodium nitrite), spices, gelling agents, acidity regulators, water, starch, soy protein, milk protein, fibre, and sugar [4].

It has been demonstrated on numerous occasions that döner kebab may present several issues from a health and hygiene perspective due to the nature of the raw materials, to the traditional cooking method used, and to the way in which it is sold and consumed, typically within small businesses run by potentially not appropriately trained staff [2,3,4]. Critical issues can therefore be identified in raw materials, human handling, environment, equipment, and production technologies.

Cooking has the effect of reducing the microbial population in döner kebab. Haskaraca and Kolsarici [5] demonstrated that cooking reduced the total mesophilic aerobic bacteria load from 6.59 log CFU/g to 4.02 log CFU/g. However, the load increased during storage under refrigerated conditions, reaching values of 6 log CFU/g after 9 days in the case of air-packaging and after 12 days for packaging in modified atmosphere and for vacuum-packaging. According to their study, there was no significant growth of coliforms or *Escherichia coli*, *Clostridium perfringens*, *Listeria monocytogenes*, or *Staphylococcus aureus*. Bingöl et al. [3] studied the effects of diverse packaging methods on the microbial quality of cooked and frozen döner kebab. Vacuum-packaging has been demonstrated to preserve the stability of the product for a period of 12 months. A significant reduction in microbial population was observed, to a greater extent in the vacuum-packed product than in the air-packed one. Specifically, a reduction of 1.6 log CFU/g was recorded in the vacuum-packed sample, whereas in the air-packed sample, a reduction of 0.68 log CFU/g was observed over the same period. According to the sensory evaluation, the air-packed product could be considered acceptable up to 6 months, while the vacuum-packaged product was found to be acceptable up to 12 months, although around the ninth month, a fat leakage was noted. Other authors, studying the microbiological quality of döner kebab, highlighted health and hygiene problems due to the presence of pathogenic microorganisms, such as *Listeria* spp. and *Salmonella* spp., which were detected in both raw and cooked kebabs [2,5,6,7,8,9,10,11,12]. *L. monocytogenes* has been found on the product, in the environment, in the equipment of the production plant, and in sales rooms [10]. Insufficient hygienic preparation of operators, no use of disinfectants, and inappropriate thawing of kebab cones performed at room temperature or during the cooking itself were the major causes. In addition to aspects strictly related to the protection of consumers’ health, it is also important to consider factors related to the deterioration of meat products during the assessment of their quality. The main mechanisms underlying the degradation of meat products are microbial alteration, enzymatic autolysis, lipid oxidation, and protein oxidation [13]. The TVB-N, resulting from protein and amino acid degradation, and the malondialdehyde index, resulting from fat oxidation, were used as indicators of the quality degradation. The amount of post-mortem TVB-N is directly related to degradative activities of microbiological and enzymatic origin, which is why it is used as a parameter to assess the quality of meat products [14,15,16]. The döner kebab can be generally considered a food product high in fat, with content ranging from 14% to more than 28% according to Vazgecer et al. [7] and Nemati et al. [8], and therefore it may be susceptible to degradation processes induced by lipid oxidation, with the consequence of deterioration in terms of nutritional quality, flavour, texture, and appearance, reducing shelf-life and causing economic losses. The malondialdehyde index can be determined experimentally and provides information about the degree of oxidation of a food product. Consequently, this parameter was measured during the shelf-life of the kebab under investigation [17,18,19,20]. Given the potential safety concerns associated with döner kebab, it was deemed necessary to evaluate the hygienic safety, microbiological and physico-chemical quality, and stability over time of a product comprising both raw and cooked döner kebab, air-packaged and stored under refrigerated and frozen conditions. An additional study involved a challenge test to assess the behaviour of *Listeria monocytogenes* in kebab stored at 4 ± 2 °C and −18 °C

## 2. Material and Methods

A total of 12 batches of raw and cooked kebabs produced in a medium-sized company in the North-East of Italy and marketed in Italy and Germany were examined. The recipe included the following: chicken meat (90%), soy, salt, flavour enhancer E621, acidifier E262, E331, dextrose, spices (celery, mustard), stabilisers E450, E451, modified potato starch, maltodextrin, vegetable broth (protein, vegetable hydrolysates, corn, salt), cellulose E460, flavourings, spice extracts. The meat and ingredients were mixed together, then the cone was formed and subsequently cooked, sliced, and air-packaged.

Three batches of raw kebab, each consisting of 100 samples, were collected and subjected to microbiological and physico-chemical analyses to assess their hygienic quality. Subsequently, three additional batches were collected and analysed to assess the hygienic and microbiological quality of cooked, refrigerated, air-packed kebabs stored for the duration of their 30-day shelf life. Specifically, the samples were stored at 4 ± 2 °C for the first 10 days and at 8 ± 2 °C for the subsequent 20 days to simulate potential thermal abuse during the marketing period. Samples were analysed at 0, 5, 10, 15, 20, 25, and 30 days. Analyses were performed in triplicate at each sampling point per each batch.

Finally, to evaluate a different storage method after the cooking process, three other supplementary batches were subjected to freezing and stored at −18 °C for 30 days. The samples were analysed at 0, 10, 20, and 30 days. Analyses were performed in duplicate per each sampling point per each batch.

The challenge test was performed to assess the behaviour of *Listeria monocytogenes* during storage in both refrigerated and frozen conditions. In the first case, the temperature was set at 4 ± 2 °C for 10 days, followed by 20 days at 8 ± 2 °C; in the second case, the temperature was set at –18 °C for 6 months. This trial was replicated three times using three different batches of cooked kebabs for each storage temperature. The products were inoculated with a mix of different *L. monocytogenes* strains before packaging and analysed at different sampling points times. Physico-chemical analyses were carried out: water activity (a_w_) measurement; pH measurement; evaluation of protein, fat, moisture, and salt content; determination of total volatile basic nitrogen; determination of malondialdehyde index, as described below.

### 2.1. Evaluation of Microbial Quality of Cooked, Air-Packaged Kebabs, Marketed Refrigerated and Frozen

The following microbiological analyses were performed: total viable count (TVC) evaluated on Plate Count Agar (PCA, Oxoid, Milan, Italy), incubated at 30 °C for 48–72 h; lactic acid bacteria (LAB) on deMan Ragosa Sharpe agar (Oxoid, Italy), double layer method, incubated at 30 °C for 48 h; total coliforms and *Escherichia coli* on Violet Bile Lactose Agar (VRBLA, Oxoid, Italy), double layer method, incubated for 24–48 h at 37° C and 44 °C, respectively; Enterobacteriaceae on Violet Bile Glucose Agar (VRBGA, Oxoid, Italy), double layer method, incubated at 37 °C for 48 h; sulphide-reducing Clostridia on Differential Reinforced Clostridial Medium (DRCM, VWR, Radnor, PA, USA) incubated at 37 °C for 24–48 h in an anaerobic jar with an anaerobic kit (gas pack anaerobic system, BBL, Becton Dickinson, Franklin Lakes, NJ, USA); Coagulase-Positive Cocci (CPC) on Baird-Parker agar medium (BP, Oxoid, Italy) supplemented with egg yolk tellurite emulsion (Oxoid, Italy) incubated at 35 °C for 24–48 h and confirmed by a coagulase test; Yeasts and Moulds on Malt Extract Agar incubated at 25 °C; *Listeria monocytogenes* and *Salmonella* spp. according to ISO [21] and ISO [22] methods, respectively.

### 2.2. Challenge Test of Air-Packaged Kebabs, Produced and Marketed Chilled or Frozen, Intentionally Inoculated with L. monocytogenes Strains

The inoculum consisted of 3 strains of *Listeria monocytogenes* from International Collections and from the Department of Agricultural, Food, Environmental and Animal Science of the University of Udine (Di4a) Collection. In particular, the following strains were used: *L. monocytogenes* NCTC 10.887 (serotype 1/2b), *L. monocytogenes* 9Di4a (serotype 4b) from meat matrix and *L. monocytogenes* 11Di4a of human origin responsible for infective listeriosis. The individual suspensions were prepared from pure colonies of each strain after growing for 3 days at 6 ± 2 °C in Plate Count Agar (Oxoid, Rodano, Italy) with added peptone water (Peptone 1 g; NaCl 30 g; distilled H_2_O 1000 mL; a_w_ 0.97). The suspensions were standardized at an optical density (OD_600_) of 0.1, corresponding to 10^7^ CFU/mL. Then, a mother suspension was prepared mixing together the suspensions containing the different types of *L. monocytogenes* in a 1:1:1 ratio in peptone water (NaCl 3%; a_w_ 0.97; log 7 CFU/mL).

An inoculum of mother suspension was added in the kebab to obtain a final inoculation value of approximately 10^6^ UFC/g. For each of the three batches tested, approximately 42 packages were produced, each containing approximately 20 g of kebab, air-packaged using food-grade plastic film (PET/PE/EVOH/PE). The inoculated samples were left for 2 h at room temperature to promote adhesion of the micro-organisms to the meat pieces. Subsequently, 21 packages were placed in the freezer at −18 °C and analysed in 3 replicates at 0, 1, 2, 3, 4, 5, and 6 months, and 21 packages were placed in the refrigerator at 4 °C for 10 days and at 8 °C for 20 days in order to simulate possible thermal abuse and analysed at 0, 5, 10, 15, 20, 25, and 30 days. This series of analyses also included 3 replicates each. A further 42 packages were produced, obtained in the same way as described above but without *L. monocytogenes* inoculation, to be used for physico-chemical analysis.

At each sampling point, each package was initially diluted with peptone water (peptone 1 g (Oxoid, Italy); NaCl 7 g; distilled H_2_O 1000 mL) in a 1/1 ratio in Stomacher bags. After homogenisation for 12 min in Stomacher (PBI, Italy), decimal dilutions were prepared. LAB, TVC, and *Listeria monocytogenes* were determined as described in Section 2.1.

### 2.3. Physico-Chemical Analysis

Water activity (a_w_) was assessed by using an AquaLab water activity meter (Decagon, San Francisco, CA, USA). The pH value was measured using a pH meter (Basic 20, Crison Instruments, Barcelona, Spain) by inserting the probe into 3 different points on each sample. Protein, fat, moisture and salt were evaluated using the methods reported in AOAC [23]. Total volatile basic nitrogen (TVB-N) was determined as described by Pearson [24]. The malondialdehyde index was evaluated via Thiobarbituric Acid-Reactive Substances (TBARS), according to the method reported by Ke et al. [25]. Data were interpreted through the proposal of several authors [25,26]. In summary, foods can be considered not rancid when TBARS values are <8 nmol malondialdehyde/g product, slightly rancid when TBARS values are between 9 and 20 nmol malondialdehyde/g product, and rancid/non-acceptable when TBARS is >21 nmol malondialdehyde/g product. Briefly, the total volatile basic nitrogen (TVB-N) was estimated by boiling a mix of distilled water (50 mL) and 10 g product in the presence of MgO (25 mL, 2% *w*/*v*). The distillate was collected in a solution of boric acid and titrated with sulfuric acid in the presence of methyl red. Thiobarbituric acid value (TBARS) was determined directly by spectrophotometric quantification of compounds obtained by the distillation of a mix consisting of distilled water (50 mL) and meat product (10 g), acidified with hydrochloric acid (2.5 mL, 4 N) until reaching pH 1.5. Then, 5 mL of the distillate was treated with 5 mL of a solution of thiobarbituric acid (TBA), obtained by mixing TBA (0.2883) in acetic acid (90%) placed in boiled water for 35 min. After cooling, the solution was read at 538 nm. Three analyses were performed at each sampling point.

### 2.4. Sensory Analysis

Sensorial analyses were performed by 20 non-professional and non-trained assessors (10 women and 10 men, representing food technology students aged between 22 and 24 years of age). The choice of non-professional tasters was mandatory because they represent typical consumers. Sensory analysis was performed based on the triangle test [27]. Three lots of döner kebab, each represented by 10 samples, were evaluated by tasters who were asked to evaluate the influence of the time of the refrigerated storage (0 days versus 30 days) on product quality. In short, three samples, two of which were identical, coded with three-digit numbers, were given in randomized service order, and the assessors were asked to find the different ones. Each tested sample was cooked at 120 °C for 15 min in an oven. After cooking, the products were cooled at 65 °C and samples were presented, wrapped in aluminium foil, in a quiet room, and the answers were collected on a paper card. The assessors who identified the different samples were asked to indicate their preference. The scoring system used was (0 days versus 30 days): 1 (excellent), 2 (good), 3 (sufficient), and 4 (scarce). Statistical evaluation of the results was carried out according to Stone and Sidel [28].

### 2.5. Statistical Analysis

Data obtained from microbiological and physico-chemical analyses of the kebab samples were compared using statistical analysis: analysis of variance (One Way ANOVA), averages separated using Tukey’s range test.

## 3. Results and Discussion

### 3.1. Microbiological Counts of Raw Döner Kebab

As highlighted in Table 1, the microbiological load of the analysed samples was found to be heterogeneous. TVC showed values ranging from 4.00 to 7.00 log CFU/g. Also, Enterobacteriaceae were counted with values stretching from 1 to 4 log CFU/g. Only 10% of the sample analysed enclosed *E. coli*, showing counts between 1 and 2 log CFU/g. LAB counts varied between 1 and 4 log UFC/g, while the concentration of moulds and yeast was found to be between 1 and 3 log UFC/g. Finally, the load of CNC and sulphite-reducing Clostridia was always below the detection limit of the method (<10 CFU/g). *Salmonella* spp. and *Listeria monocytogenes* were always absent in 25 g.

Considering the 100 samples analysed, a wide microbiological variability can be noticed, as already reported by other Italian authors [4,11,29]. Moreover, the microbial contamination of the kebab is conditioned by numerous factors, common to every processed food, including the hygienic characteristics of the raw material, the storage phases, the hygienic conditions of the staff, as well as the workplace and equipment [4].

In this case, the kebab appeared to be of modest hygienic quality, with 60 samples presenting total bacterial counts of more than 6 log CFU/g. This was probably due to the use of poultry meat, which is usually more contaminated than pork and beef [4]. This is confirmed by the presence of Enterobacteriaceae and *E. coli* observed in the examined samples, albeit in low concentrations. However, the results are similar to those found by other authors [6,7,9,30,31], especially regarding TVC and Enterobacteriaceae. In contrast to those studies, *Salmonella*, *Staphylococcus aureus*, Clostridia H_2_S^+^ and *L. monocytogenes* were not detected, despite some authors having demonstrated that *Salmonella* spp. and *E. coli* can be particularly present on areas of broiler carcasses, and some microorganisms such as *Salmonella* spp. may attach to poultry skin and may be difficult to remove [32]. A great risk could arise from the presence of these pathogens when chicken skin is added to chicken doner kebab [9]. For these reasons, some manufactures recommend that the skin used in döner kebabs must be heated in order to increase the chemical and microbiological quality of chicken döner kebabs and to eliminate serious microbiological risk [9]. However, based on the Italian Guidelines for raw meat “Guidelines for Official Control pursuant to REG. Ce 882/04 and 854/04 reported by the State Regions Conference of 10/11/2016”, the tested products can be considered acceptable. Indeed, the guidelines indicate that raw meat with TVC concentrations below 7 log CFU/g can be considered acceptable and of adequate quality. In general, microbial loads are higher in raw samples than in cooked ones. The subsequent heat treatment could reduce microbial loads by 3 or 4 log CFU/g [4,8,9]. Cooking generally results in a reduction of microbial loads; however, in several instances, it has been found to be inadequately performed, as confirmed by organoleptic evaluation. There have been reported cases in various countries where foodborne illnesses occurred, with the consumption of contaminated döner kebabs identified as a potential cause [33,34,35,36].

### 3.2. Microbiological and Physico-Chemical Quality of Air-Packaged Kebabs, Stored Under Refrigeration and Freezing Conditions for the Shelf-Life Period

The evaluation of cooked, packaged kebabs stored under refrigeration or freezing conditions is presented in Table 2 and Table 3.

*Salmonella* spp. and *Listeria monocytogenes* were absent in 25 g, and sulphite-reducing Clostridia were always below the detection limit (10 CFU/g). Throughout the refrigerated storage period at 4 ± 2 °C for 10 days followed by 20 days at 8 ± 2 °C, microbial growth was observed across all the different microbial groups under examination, as expected (*p* < 0.05) (Table 2).

The product demonstrated a superior microbiological quality compared to that observed by Ziino et al. [4] in some retail outlets in the provinces of Palermo and Messina. In their study, the authors detected TVC values of up to 6.30 log CFU/g and the presence of sulphite-reducing Clostridia in concentrations of 2.00 log CFU/g in cooked and ready-to-eat products. Compared to the observations of Haskaraca and Kolsarici [5], the initial contamination was similar, despite the fact that the product was air-packaged and stored in thermal abuse.

The product initially exhibited levels of moulds and yeasts that were comparable to those previously documented by Ismail et al. [37], but no significant growth was observed over time. Among the microbial populations observed, the LAB population showed the most evident growth (*p* < 0.05), and it was found to be the dominant population, in agreement with the observations of Patsias et al. [38], reaching a value of 8.9 ± 0.2 log CFU/g at day 30. Contrary to the findings of the same authors, growth of Enterobacteriaceae was also observed.

Finally, no significant change in pH was observed over time (*p* > 0.05) (Table 2), and the values are comparable to those observed by several authors [3,4,6,7]. The pH values satisfy the Standards and Industrial Research of Iran, recommending pH values ranging from 5.8 to 6.2 for kebabs [39]. However, pH values of these products may vary since it depends on the materials used in marinating, temperature, and storage time. It has been observed that when a mixture of onions and spices is used, pH values drop below 5.8 units [40,41].

The results are similar to those found by several authors [4,6,7,8] for cooked döner samples. Comparing Table 1 and Table 2, it is clear that the raw döner samples had significantly higher microbial concentrations than the cooked ones. As expected, the cooking process had the effect of decreasing the microbial counts by several logs CFU/g, as already demonstrated by Nemati et al. [8], who observed a decrease of about 3.5 log CFU/g in the TVC in cooked samples in comparison to raw ones. The same reduction was observed for *Clostridium perfringens* and for moulds and yeasts [8]. In addition, CPC were also found in raw meat by the same authors, whereas they were absent in the cooked samples. Additionally, other authors observed a 3 log CFU/g reduction in TVC after the cooking process [5,6,42,43,44]. The critical total microbial limit for cooked meat and meat products has been reported as 6 log CFU/g [5,30,31]. After this value, the spoilage begins, and it is stated that the microbial load could cause sensory perceptible odour when 8 log CFU/g values are reached [31].

In this study, the limit of 6 log CFU/g in TVC was reached between day 20 and day 30, despite the fact that the meat contained sodium acetate, an additive used to control microbial growth, which was supposed to enhance stability. It should be noted that the limit was proposed for products stored at 4 ± 2 °C and not under thermal abuse, as was planned. However, the product under examination exhibited no incipient signs of spoilage as demonstrated by the TVB-N values and the malondialdehyde index, which, at the end of the shelf-life, were recorded at 32.2 mg N/100 g and 1.4 nmol MDA/g (*p* < 0.05), respectively. It can thus be concluded that, when stored under refrigerated conditions, the shelf-life of the kebab is about 30 days, in accordance with the limit indicated by Commission Regulation (EC) No. 2074/2005 [45], which sets the maximum TVB-N value for acceptability at 35 mg N/100 g. However, the data regarding acceptability related to the concentration of TVB-N are conflicting and vary according to the author. TVB-N values of 25–28 mg N/100 g have been suggested for fresh chicken meat as a level above which the product becomes unacceptable from a sensory point of view [14,15,46,47]. On the other hand, Balamatsia et al. [48] demonstrated and proposed limit values of 40 mg N/100 g for chicken meat. In our opinion, values of 25–28 mg N/100 g are too reductive for the examined product. If this limit had been applied, the shelf-life would have been only 10 days, whereas on day 10, the product was indeed acceptable from a microbiological and sensory point of view.

The kebab consisted of processed meat, using mild technologies. No critical limit has been established for this kind of product, excluding the ones cited. However, applying the limits for fish cited in EC Reg. 2074/2005 and the conclusions obtained by Balamatsia et al. [48], and considering that the product under examination did not present abnormal odours, there is no doubt that its acceptability can be extended to 30 days.

Considering the malondialdehyde index, the product can be considered acceptable for up to 30 days. The TBARS value never exceeded 8 nmol MDA/g product. Therefore, the refrigerated kebab under study can be classified as ‘non-rancid’ according to Che Man and Ramadas [26] and Ke et al. [25]. The malondialdehyde index, however, increased during storage, probably due to air which was present in the packaging. It could be assumed that Modified Atmosphere Packaging (e.g., 30% CO_2_–70% N_2_) without air could limit or reduce fat oxidation, as observed by Patsias et al. [38].

*Salmonella* spp. and *Listeria monocytogenes* were absent in 25 g, while sulphite-reducing Clostridia were always below the detection limit.

Regarding the frozen product, as expected, no significant changes in microbial growth were observed over time (*p* > 0.05). However, significant differences were observed between the various samples, showing that the microbial quality of the product was not homogeneous.

Similar results were obtained by Bingöl et al. [3] in a work aimed at assessing the quality of air-packaged (AP) and VP frozen kebabs, using barrier foam trays (LLDPE/EVOH multilayer), wrapped with a polyethylene film for air-packaging (AP) or heat-sealed with a Multivac packaging unit for VP with multilayer polyolefin with PVDC film. Comparing our results to the above authors, this study indicates that the freezing process not only prevents microbial growth but also produces a reduction in microbial counts over time. According to Bingöl et al. [3], the reduction is higher in VP kebabs than in air-packaged ones, and the TVC values of the product were significantly different (*p* < 0,05) between the product packaged in air and the VP ones. In addition, they showed that storage time and packaging conditions were significant factors influencing TBARS values. No influence was attributed to the films. The TBARS values of all samples increased in up to 8 months of storage time, then decreased until the end of storage; levels in VP samples remained lower than those of AP ones during the entire storage time. It was observed that the evolution of microorganisms was not attributed to the film but rather to the types of packaging utilised. The SVP process was observed to ensure the microbiological safety of döner kebab products for a period of at least 100 days while stored at 4 °C. SVP was the most effective technique among the investigated packaging methods for increasing the shelf-life during cold storage without adding any preservatives or additives and protected the microbial quality of döner kebabs 20, 6, and 5 times more than AP, MAP, and VP, respectively [5].

Freezing, as expected, also inhibited the production of TVB-N and MDA in the investigated samples. The values observed at the end of the 30 days were largely acceptable (Table 3).

### 3.3. Challenge Test Performed on Air-Packaged Kebabs, Stored Under Refrigeration and Freezing Conditions, Intentionally Inoculated with Listeria monocytogenes Strains

Data on total viable count, lactic acid bacteria, *Listeria monocytogenes*, and pH during the challenge test are given in Table 4 and Table 5.

The challenge test showed no growth of *L. monocytogenes* during storage under refrigerated conditions. Contamination indeed decreased by 0.5 log CFU/g (*p* < 0.05). In addition, data show that, although air-packaging was still used, microbial growth was slower during these tests than in previous ones, regarding both TVC and LAB. There was also a slight decrease in pH value (*p* < 0.05). It is likely that the development of competitor bacteria prevented the development of *Listeria* spp. and *L. monocytogenes* [5,49,50,51]. These results concur with those obtained by Haskaraca and Kolsarici [5], who, in a work meant to evaluate the development of different microorganisms naturally present in cooked kebabs, noted the lack of growth of *L. monocytogenes* over time. It can be hypothesized that the development of LAB and the consequent reduction in pH are responsible for the failure to develop the *L. monocytogenes* intentionally added to the considered doner kebab samples. It is well known that the development of LAB naturally present in meat or intentionally added as bioprotective starters prevents the development of pathogens and *L. monocytogenes* in particular [50,51]. Furthermore, LAB can decrease the concentration of the pathogen by 2–3 log CFU/g, but not eliminate it completely (Tolerance 0), although LAB produces lactic acid and sometimes also bacteriocins [50,52,53,54]. However, it is possible to permanently eliminate it from the meat used for its production or directly from the finished product via good cooking technology and, in the case of purchasing a refrigerated or frozen product, with a thermal heating treatment carried out before consumption [10,55].

During storage at −18 °C, as expected, there was no microbial growth. TVC decreased from 4.4 ± 0.1 log CFU/g to 3.5 ± 0.2 CFU/g (*p* < 0.05). LAB decreased from 3.8 ± 0.2 log CFU/g to 3.5 ± 0.3 CFU/g (*p* < 0.05). *Listeria monocytogenes* count decreased by 1 log CFU/g (*p* < 0.05), while the pH remained stable (Table 5).

The presence of pathogens in kebabs represents a real risk for the consumer, especially in case of inadequate application of hygiene standards throughout the production and storage phase. Greater risks are linked to doner kebabs produced at the catering and take-away restaurant level. In Italy, a first case of food poisoning due to staphylococcal enterotoxin type B in döner kebab has already been described [56].

Consequently, government agencies should accelerate to establish projects regarding food safety regulations of traditional foods towards qualifying for whole membership in the European Union. It is necessary to standardize the food safety and processing of the majority of traditional foods, like doner kebab, produced at industrial and artisanal levels since the small food processing plants and restaurants producing traditional foods still can possibly ignore sanitation and hygiene rules, which might create risk for public health [57]. Catering foods distributed to several places are consumed by large numbers of people, so the presence of toxin-producing bacteria and *Listeria* spp. in RTE food is vital for public health [52]. For these reasons, regular training of employees on hygiene practices is essential, and monitoring of the results should be ensured to mitigate future risks. As a result, it is essential to implement good hygienic and manufacturing practices within the catering industry to prevent microbiological contamination and ensure food safety for consumers.

### 3.4. Physico-Chemical Composition of Cooked Kebab

The chemical and physical composition of the cooked kebab (Table 6) under examination is relatively similar to one of the other kebab products observed in the literature [6,7]. There are a few differences in terms of moisture, which was always below 55% in this case, and protein, which reached 28,8% (Table 6). Literature data report average values of moisture of 61%, of protein 17%, and of fat 17% [8]. The fat concentration observed in our samples was between 19–20%, which is in agreement with the Institute of Standards and Industrial Research of Iran, which suggested a maximum fat content of 20% for kebabs [39].

No significant changes in the chemical composition of the product were observed during the storage period, either under refrigerated conditions for 30 days or at −18 °C for 6 months. In both cases, the döner kebab could be considered ‘not rancid’, as measured by the malondialdehyde index. For the refrigerated product packed in air, the TBARS value reached 0.9 ± 0.3 nmol MDA/g after 30 days, while for the challenge product, a value of 0.3 ± 0.2 nmol MDA/g was reached during the same period. Regarding the frozen product, the TBARS value reached 1.2 ± 0.1 nmol MDA/g after only 30 days in the first test, while in the second test (challenge), the TBARS value was 0.7 ± 0.2 nmol MDA/g after 6 months. This might be due to a different concentration of unsaturated fats and to the fact that freezing may have slowed down oxidation reactions [58]. These MDA values are satisfactory considering that both frozen and refrigerated products were packed in air and thus potentially more prone to oxidation.

Storage time and packaging conditions have been proven to be the most significant factors contributing to modifications in TBARS values [3], as confirmed by Gonulalan et al. [59], who reported that, over 60 days of storage time, TBARS values of frozen sucuk döner increased, fat content increased as a result of moisture content loss, and pH values decreased. Even if TBARS values always tend to increase during storage time, vacuum-packaging may offer some protection and slow down the phenomenon since, as observed by Bingöl et al. [3] and Ergonül and Kundakci [60], TBARS levels in vacuum-packaged samples tend to remain lower than those in air-packaged ones.

TVB-N values increased significantly at both temperature conditions (*p* < 0.05). However, for both refrigerated and frozen products, the levels of TVB-N can be considered acceptable, being <35 mg N/100 g. Specifically, a mean value of 19.4 mg N/100 g was observed in the frozen product after 6 months and a value of 32.5 mg N/100 g was found in the refrigerated product at day 30 (Table 6).

TVB-N concentration in the frozen product largely complies with the limits proposed by several authors [14,15,46,47]. Regarding the refrigerated product, the final value of 32.5 mg N/100 g may or may not be considered acceptable, since different authors propose different critical limits. However, as mentioned above, it can be stated that in this case, critical limit values of 40 mg N/100 g can be applied for TVB-N [48]. Therefore, it can be stated that this döner kebab can be stored at 4 ± 2 °C for about 30 days and at −18 °C for 6 months.

The work demonstrates that döner kebab can be healthy, but only when applying good manufacturing practices. The results of the analysis indicate that it is possible to produce it in the absence of pathogenic microorganisms. In addition, the packaging and the storage temperatures limited or blocked microbial growth. Therefore, it can be stated that this döner kebab can be stored at 4 ± 2 °C for about 30 days and at −18 °C for 6 months. However, in particular regarding the frozen product, the shelf-life period may be increased, considering the TVB-N value and the MDA index observed after 6 months.

### 3.5. Sensorial Analysis

The twenty nonprofessional subjects were unable to distinguish the two types of döner kebab (0 and 30 days storage). The triangle test methodology demonstrated that the refrigerated storage for 30 days did not influence the odour or flavour of the döner kebab. Thus, they established that there was no difference between the two samples. This analysis confirmed the microbial and physico-chemical results and the acceptability of the product after up to 30 days of refrigerated storage.

## 4. Conclusions

Döner kebab is a food product which is widely spread in Italy and in the European Union. Several studies since the 1980s have shown that it could represent a potential risk for consumers’ health, especially when it is not consumed immediately after cooking. Cooking has the effect of reducing the microbial load but not of eliminating it completely. Kebab has chemical and physical characteristics which can allow microbial growth during storage, both regarding spoilage and potentially pathogenic microorganisms. The study showed that the kebab under investigation, stored under refrigeration and considering possible thermal abuse, can be considered safe and healthy for a period of 30 days, especially from a microbiological point of view. The problem is related to the microbial or tissue activity resulting in a possible increase in TVB-N concentration. However, although some authors set the limit at 25.5 mg N/100 g, a limit value of 40 mg N/100 g can still be accepted, as reported for certain categories of fishery products and reference methods (EC Reg. 2074/2005: limit values and reference methods). The same product, frozen and stored at −18 °C, can be considered stable for at least 6 months, both from a microbiological and a physico-chemical point of view. Neither the chilled nor the frozen product showed problems regarding fat oxidation during the shelf-life period. The challenge test showed that the product stored at both 4 ± 2 °C for 10 days and 8 ± 2 °C for 20 days, as well as at −18 °C for 6 months, did not allow significant growth of *Listeria monocytogenes*.

Air-packaging has been proven to be beneficial to the preservation of the product and maintained its microbiological and physico-chemical quality at an adequate level.

Despite the good results and in order to increase its shelf-life, reducing the microbial growth in the refrigerated product, future directions could be to investigate different plastic films and packaging such as MAP, VP, and SVP.

## Figures and Tables

**Table 1 microorganisms-13-00701-t001:** Microbial load of 100 samples of raw kebab.

Microorganisms	Percentage of Samples with Count CFUC/g						
	<10	10^1^–10^2^	10^2^–10^3^	10^3^–10^4^	10^4^–10^5^	10^5^–10^6^	10^6^–10^7^
TVC					10	30	60
Enterobacteriaceae		45	35	20			
Total coliforms		45	40	15			
*Escherichia coli*	90	10					
Clostridia H_2_S^+^	100						
Lactic Acid Bacteria		60	20	20			
Moulds and Yeasts		90	10				
CPC	100						
*Salmonella* spp.	Absent *						
*Listeria monocytogenes*	100						

Legend: * Absent/25 g product; TVC: total viable count. CPC: Coagulase-Positive cocci.

**Table 2 microorganisms-13-00701-t002:** Microbiological and physico-chemical evolution of air-packaged kebab stored at 4 ± 2 °C * for 10 days and at 8 ± 2 °C for 20 days.

Parameters	Time (Days)						
	0 *	5 *	10 *	15	20	25	30
TVC	4.5 ± 0.2 a	4.7 ± 0.1 a	4.5 ± 0.2 a	5.3 ± 0.2 b	5.8 ± 0.1 c	6.0 ± 0.3 c	6.2 ± 0.1 c
Lactic Acid Bacteria	3.1 ± 0.2 a	3.1 ± 0.2 a	3.5 ± 0.1 b	8.0 ± 0.1 c	8.6 ± 0.8 c	8.6 ± 0.8 c	8.9 ± 0.2 c
Moulds and Yeasts	2.3 ± 0.2 a	2.4 ± 0.1 a	2.3 ± 0.1 a	3.0 ± 0.4 b	3.2 ± 0.2 b	3.2 ± 0.3 b	3.5 ± 0.4 b
Total coliforms	1.8 ± 0.1 a	1.8 ± 0.3 a	1.6 ± 0.2 a	1.6 ± 0.1 a	2.8 ± 0.3 b	3.2 ± 0.1 b	3.6 ± 0.5 b
Enterobacteriaceae	1.9 ± 0.2 a	1.8 ± 0.1 a	1.9 ± 0.2 a	1.9 ± 0.2 a	2.9 ± 0.1 b	3.4 ± 0.1 c	3.7 ± 0.2 c
pH	6.6 ± 0.1 a	6.5 ± 0.2 a	6.3 ± 0.4 a	6.2 ± 0.3 a	6.1 ± 0.4 a	5.9 ± 0.2 a	5.8 ± 0.3 a
MDA nmol/g	0.6 ± 0.2 a	0.9 ± 0.3 a	1.2 ± 0.1 a	1.2 ± 0.1 a	1.4 ± 0.1 b	1.4 ± 0.1 b	1.4 ± 0.2 c
TVB-N mg N/100 g	15.0 ± 3.0 a	19.4 ± 6.2 a	29.3 ± 1.5 b	29.5 ± 1.2 b	32.2 ± 2.1 b	32.3 ± 2.2 b	32.5 ± 2.3 b

Data (log CFU/g) indicate means ± standard deviation. Means with the same letters within the same rows are not significantly different (*p* < 0.05). TVC: Total Viable Count; TVB-N: Total Volatile Basic Nitrogen; MDA: Malondialdehyde.

**Table 3 microorganisms-13-00701-t003:** Microbiological and physico-chemical evolution of air-packaged kebab stored at −18 °C for 30 days.

Parameters	Time (Days)			
	0	10	20	30
TVC	4.2 ± 0.1 a	4.7 ± 0.1 b	4.5± 0.2 b	4.3 ± 0.1 a
Lactic Acid Bacteria	3.9 ± 0.2 a	4.0 ± 0.1 a	4.0 ± 0.1 a	3.2 ± 0.6 a
Moulds and Yeasts	3.7 ± 0.1 a	3.4 ± 0.2 a	3.9 ± 0.2 ab	3.3 ± 0.3 a
Total coliforms	2.4 ± 0.1 a	1.9 ± 0.1 b	2.4 ± 0.2 a	1.7 ± 0.1 b
Enterobacteriaceae	2.2 ± 0.1 a	2.2 ± 0.1 a	3.1 ± 0.1 b	1.7 ± 0.2 c
pH	6.4 ± 0.1 a	6.3 ± 0.2 a	6.3 ± 0.2 a	6.2 ± 0.1 a
MDA nmol/g	0.6 ± 0.1 a	0.9 ± 0.1 a	1.2 ± 0.2 a	1.2 ± 0.1 a
TVB-N mg N/100 g	15.0 ± 2.0 a	18.6 ± 3.0 a	18.3 ± 1.1 a	19.1 ± 1.0 a

Data (log CFU/g) indicate means ± standard deviation. Means with the same letters within the same rows are not significantly different (*p* < 0.05). TVC: Total Viable Count; TVB-N: Total Volatile Basic Nitrogen; MDA: Malondialdehyde.

**Table 4 microorganisms-13-00701-t004:** Evolution of TVC, LAB, *Listeria monocytogenes*, and pH in kebab samples stored at 4 ± 2 °C * for 10 days and at 8 ± 2 °C for 20 days.

Parameters	Time (Days)						
	0 *	5 *	10 *	15	20	25	30
TVC	4.4 ± 0.1 a	4.3 ± 0.2 a	4.4 ± 0.1 a	4.5 ± 0.2 a	4.6 ± 0.1 a	4.8 ± 0.1 b	5.0 ± 0.1 b
Lactic acid bacteria	3.8 ± 0.2 a	3.9 ± 0.1 a	4.0 ± 0.2 a	4.2 ± 0.1 a	5.2 ± 0.3 b	5.6 ± 0.1 c	6.2 ± 0.2 d
*Listeria monocytogenes*	5.9 ± 0.1 a	5.9 ± 0.1 a	5.8 ± 0.2 a	5.6 ± 0.1 b	5.5 ± 0.2 b	5.3 ± 0.2 b	5.4 ± 0.1 b
pH	6.1 ± 0.1 a	6.0 ± 0.2 a	5.9 ± 0.1 a	5.8 ± 0.1 a	5.8 ± 0.1 a	5.7 ± 0.2 b	5.7 ± 0.2 b

Data (log CFU/g) indicate means ± standard deviation. Means with the same letters within the same rows are not significantly different (*p* < 0.05). TVC: Total Viable Count.

**Table 5 microorganisms-13-00701-t005:** Evolution of TVC, LAB, *Listeria monocytogenes*, and pH in kebab samples stored at −18 °C for 6 months.

Parameters	Time (Months)						
	0	1	2	3	4	5	6
TVC	4.4 ± 0.1 a	4.4 ± 0.2 a	4.0 ± 0.1 b	4.0 ± 0.2 a	3.9 ± 0.1 a	3.7 ± 0.1 c	3.5 ± 0.2 c
Lactic acid bacteria	3.8 ± 0.2 a	3.7 ± 0.2 a	3.8 ± 0.1 a	3.8 ± 0.1 a	3.5 ± 0.1 b	3.6 ± 0.1 b	3.5 ± 0.3 b
*Listeria monocytogenes*	5.9 ± 0.1 a	5.9 ± 0.2 a	5.5 ± 0.1 b	5.4 ± 0.2 b	5.2 ± 0.1 b	5.1 ± 0.2 b	4.9 ± 0.1 c
pH	6.0 ± 0.1 a	6.0 ± 0.2 a	5.9 ± 0.1 a	5.9 ± 0.1 a	5.9 ± 0.1 a	5.8 ± 0.2 a	5.9 ± 0.1 a

Data (log CFU/g) indicate means ± standard deviation. Means with the same letters within the same rows are not significantly different (*p* < 0.05). TVC: Total Viable Count.

**Table 6 microorganisms-13-00701-t006:** Evolution of physico-chemical characteristics of kebab inoculated with *L. monocytogenes* over time.

Physico-Chemical Characteristics	Type of Product			
	Refrigerated at 4 ± 2 °C for 10 Days and at 8 ± 2 °C for 20 Days		Frozen at—18 °C for 6 Months	
	0	30	0	6
Moisture %	53.5 ± 3.5 a	51.5 ± 4.2 a	53.5 ± 3.5 a	52.3 ± 2.5 a
Protein %	25.5 ± 2.0 a	26.5 ± 1.0 a	25.5 ± 2.0 a	26.0 ± 2.8 a
Fat %	19.3 ± 1.8 a	20.5 ± 2.3 a	19.3 ± 1.8 a	20.1 ± 1.4 a
Salt %	1.1 ± 0.3 a	1.3 ± 0.3 a	1.1 ± 0.3 a	1.2 ± 0.2 a
MDA nmol/g	0.1 ± 0.1 a	0.3 ± 0.2 a	0.1 ± 0.1 a	0.7 ± 0.2 a
TVB-N mg N/100 g	15.0 ± 3.0 a	32.5 ± 2.3 c	14.7 ± 2.6 a	19.4 ± 2.7 b
a_w_	0.971 ± 0.01 a	0.970 ± 0.01 a	0.971 ± 0.01 a	0.971 ± 0.02 a

Means with the same letters within the same lines are not significantly different (*p* < 0.05); TVB-N: Total Volatile Basic Nitrogen; MDA: Malondialdehyde.

## Data Availability

The original contributions presented in this study are included in the article. Further inquiries can be directed to the corresponding authors.

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
