# Peer review of "Hygienic Quality of Air-Packed and Refrigerated or Frozen Stored Döner Kebab and Evaluation of the Growth of Intentionally Inoculated Listeria monocytogenes"

_microorganisms, 2025, doi:10.3390/microorganisms13040701_

Round 1

Reviewer 1 Report

Comments and Suggestions for Authors

The study conducted by Coppola et al. addresses a relevant topic on the hygienic quality of air-packed and refrigerated or frozen stored Döner Kebab and evaluation of the growth of Listeria monocytogenes intentionally inoculated. I believe after the following revisions, the manuscript can be considered for publication in Microorganisms:

In the abstract, please highlight the main results of your study (values) and indicate some directions to be followed in future research.

The Introduction is well documented, but it would be better to subdivide it into subsections for a better comprehension and some parts have no references. This is just an example: Lines 38-55.

The physical-chemical analysis has to be better explained and I suggest the inclusion of a flowchart in the Materials and Methods section with all the steps taken to perform this study-.

The Results and Discussion are fine, but some information in the Tables is in Italian. The authors should also indicate the study’s strengths and limitations before the Conclusions.

The Conclusions should be more concise and Future Perspectives are missing.

Author Response

Dear referee

Enclosed you will find a copy of our revised manuscript, “Microorganisms- 3546331” entitled “Hygienic Quality of Air-Packed and Refrigerated or Frozen Stored Döner Kebab and Evaluation of the Growth of Listeria monocytogenes Intentionally Inoculated” submitted to Microorganisms.

The authors would like to thank the reviewers for their careful reading of the manuscript and the resulting constructive comments and suggestions. Basically, we agree with all the points raised by the reviewers, and wherever possible the manuscript has been modified as recommended. All reviewer comments are in black plain font, whereas our response is described in red plain font. We have made the changes and corrections based on the reviewer’s suggestions. We evaluated the comments and prepared a point-by-point response to each one of them.

Referee 1

The study conducted by Coppola et al. addresses a relevant topic on the hygienic quality of air-packed and refrigerated or frozen stored Döner Kebab and evaluation of the growth of Listeria monocytogenes intentionally inoculated. I believe after the following revisions, the manuscript can be considered for publication in Microorganisms:

In the abstract, please highlight the main results of your study (values) and indicate some directions to be followed in future research.

Answer: Thanks - I modified it:

Lines 13-48. Döner kebab, a meat product of Middle-Eastern origin, has gained significant popularity and is now widely consumed across Europe. The recipe varies depending on the area, with beef, turkey, lamb or chicken being used as main ingredients. The aim of this work was to assess the hygienic-sanitary quality of raw and cooked döner kebabs stored at 4 ± 2 °C for 10 days and at 8 ± 2 °C for the next 20 days or frozen (- 18 °C) for one month. One additional aim was to determine the potential growth of Listeria monocytogenes intentionally inoculated in cooked döner kebab during storage at 4 ± 2 °C or frozen. The concentration of Total Viable Count (TVC) and the Enterobacteriaceae of the 100 samples of raw döner kebab were less than 7 log CFU/g and 4 log CFU/g, respectively. Consequently, the samples can be considered acceptable, and similar to traditional raw meat. The cooked döner kebab can be considered safe for a period of 30 days, especially from a microbiological point of view, when stored under refrigerated conditions, also taking into account possible thermal abuse. Coagulase Positive Cocci (CPC), Clostridium H2S+,Salmonella spp. and Listeria monocytogenes were never found in all samples. After 30 days the TVC was at level of 6 log CFU/g and Enterobacteriaceae less than 4 log CFU/g. The main concern was related to microbial or tissue activity, resulting in an increase in total volatile basic nitrogen (TVB-N) content. However, in the cooked samples, the TVB-N content remained below 40 mg N/100 g at the end of the shelf-life period (32.5 mg N/100 g), which is still considered an acceptable value. In addition, the level of Malondialdehyde (MDA) was found to be within acceptable limits, with a reading of 1.4 nmol/g attained after 30 days. The same product, when frozen and stored at -18 °C, can be considered stable for a minimum of 6 months, both from a microbiological and a physico-chemical point of view. No microbial growth was observed. The TVB-N and the MDA levels increased, but after 6 months their levels were still acceptable, with values of 19.1 mg N/100 g and 1.2 nmol/g, respectively. These observations demonstrate low protein degradation and lipid oxidation during the shelf-life period. The challenge test showed that Listeria monocytogenes did not grow in döner kebab stored at either 4 ± 2 °C for 10 days and 8 ± 2 °C for 20 days, nor when stored at -18 °C for 6 months. The concentration of L. monocytogenes was found to be 5.4 log CFU/g in the refrigerated products and 4.9 log CFU/g in the frozen products. At the end of the shelf-life period, the L. monocytogenes load in both products was lower than the initial concentration that had been added. Finally, the use of air-packaging has been proven to be beneficial to the preservation of the product and maintained its microbiological and physico-chemical properties intact. Despite the good results, the future directions could be to investigate different plastic films and packaging such as Modified Atmosphere, Vacuum and Sous Vide packaging.

The Introduction is well documented, but it would be better to subdivide it into subsections for a better comprehension and some parts have no references. This is just an example: Lines 38-55.

Answer - Thanks I reduced it

The parts from lines 38-55 were taken from the first author reference. We think it is not needed to subdivide the introduction in different sections. Consulting the literature we never found an introduction with sections. However, we think that the introduction seems to be comprehensible.

Lines 52-64 - Döner kebab, a meat product of Middle-Eastern origin, has gained significant popularity and is now widely consumed throughout Europe. The name of döner is derived from the Turkish dönmek and refers to the cooking method, which involves rotating large cones made of meat and other ingredients in open ovens, allowing them to cook slowly on the surface [1]. Döner kebab born in the 1800s in Turkey and began to spread in Europe with the great wave of Turkish immigration, starting in the 1950s. In many European countries, including Italy, a number of kebab outlets in the tens of thousands are present today. In Germany, in 2020, 400 tonnes per day of kebab were produced, generating a turnover of 4,5 billion € per year [1]. Döner kebab is typically sold and consumed in small retail outlets, where operators are used to purchasing the cones – usually in a frozen state – from industrial plants and completing the process in open ovens. The sliced meat is commonly served in sandwiches, together with a variety of vegetables and sauces, providing a convenient and inexpensive meal.

The physical-chemical analysis has to be better explained and I suggest the inclusion of a flowchart in the Materials and Methods section with all the steps taken to perform this study.

Answer – Thanks, I will explain better.

Lines 211-230 - Water activity (aw) was assessed by using AquaLab water activity meter (Decagon, USA). The pH value was measured using a pH-metre (Basic 20, Crison Instruments, Spain), by inserting the probe into 3 different points on each sample. Protein, fat, moisture and salt were evaluated using the methods reported in AOAC [23].

Total volatile basic nitrogen (TVB-N) was determined as described by Pearson [24]. The malondialdehyde index was evaluated via Thiobarbituric Acid-Reactive Substances (TBARS), according to the method reported by Ke et al. [25]. Data were interpreted through the proposal of several authors [25,26]. In summary, foods can be considered not rancid when TBARS values are < 8 nmol malondialdehyde/g product, slightly rancid when TBARS values are between 9 and 20 nmol malondialdehyde/g product, and rancid/non-acceptable when TBARS is > 21 nmol malondialdehyde/g product. Briefly: the total volatile basic nitrogen (TVB-N) was estimated by boiling a mix of distilled water (50 mL) and 10 g product in presence of MgO (25 mL, 2% w/v). The distillate was collected in a solution of boric acid and titrated with sulfuric acid in the presence of methyl red. Thiobarbituric acid value (TBARS) was determined directly by spectrophotometric quantification of compounds obtained by the distillation of a mix consisting of distilled water (50 mL) and meat product (10 g), acidified with hydrochloric acid (2.5 mL, 4 N) until pH 1.5. Then, 5 mL of the distillate was treated with 5 mL of a solution of thiobarbituric acid (TBA), obtained by mixing TBA (0.2883) in acetic acid (90%) placed in boiled water for 35 min. After cooling, the solution was read at 538 nm. Three analyses were performed at each sampling point.

It is not needed to include a flowchart, because I have just subdivided the introduction in different sections.

The Results and Discussion are fine, but some information in the Tables is in Italian. The authors should also indicate the study’s strengths and limitations before the Conclusions.

Answer – Thanks, the tables have been corrected using TVB-N in place of ABTV and all the Italian words were treated in English.

Lines 504-510 - The work demonstrates that Döner kebab can be healthy, but only applying the Good Manufacturing Practices. The results of the analysis indicate that it is possible to produce it in the absence of pathogenic microorganisms. In addition, the packaging and the storage temperatures limited or blocked microbial growth. Therefore, it can be stated that this döner kebab can be stored at 4 ± 2 °C for about 30 days and at -18 °C for 6 months. However, in particular regarding the frozen product, the shelf-life period may be increased, considering the TVB-N value and the MDA index observed after 6 months.

The Conclusions should be more concise and Future Perspectives are missing.

Answer – Thanks, I modified it.

Lines 520-542 - Döner kebab is a food product which is widely spread in Italy and in the European Union. Several studies since the 1980s have shown that it could represent a potential risk for consumers' health, especially when it is not consumed immediately after cooking. Cooking has the effect of reducing the microbial load, but not of eliminating it completely. Kebab has chemical and physical characteristics which can allow microbial growth during storage, both regarding spoilage and potentially pathogenic microorganisms.

The study showed that the kebab under investigation, stored under refrigeration, considering possible thermal abuse, can be considered safe and healthy for a period of 30 days, especially from a microbiological point of view. The problem is related to the microbial or tissue activity resulting in a possible increase in TVB-N concentration. However, although some authors set the limit at 25,5 mg N/100 g, a limit value of 40 mg N/100 g can still be accepted, as reported for certain categories of fishery products and reference methods (EC Reg. 2074/2005: limit values and reference methods).

The same product, frozen and stored at -18°C, can be considered stable for at least 6 months, both from a microbiological and a physico-chemical point of view.

Neither the chilled nor the frozen product showed problems regarding fat oxidation during shelf-life period.

The Challenge test showed that the product stored at both 4 ± 2 °C for 10 days and 8 ± 2 °C for 20 days, as well as at -18 °C for 6 months, did not allow significant growth of Listeria monocytogenes.

Air-packaging has been proved beneficial to the preservation of the product and maintained its microbiological and physico-chemical quality to an adequate level.

Despite the good results and to increase its shelf-life, reducing the microbial growth in the refrigerated product, the future directions could be to investigate different plastic films and packaging such as MAP, VP and SVP.

Reviewer 2 Report

Comments and Suggestions for Authors

The manuscript is a research article that delas with the shelf life estimation of a traditional meat product, that is kebab, under different refrigerated conditions. Challenge tests were also carried out with pathogens. The manuscript has novel insights and falls within the aimsand scope of the journal.The authors state that numerous samples were analyzed which provides a strength for the study.

The paper has been well prepared and data were treated with statistical analysis, one-way analysis of variance and Tukey's honestly significance test. This should be better defined in the relevant section. My comments deal with the packaging used.Why the authors used this multi -layer film? Did the authors run preliminary tests in other films? Another comment is that the authors should compare their data with studies using other packaging films. Pros and Cons in this context.

Finally, sensory anaysis of the treted product under refrigeration  is missing.

A revision is required.

Author Response

Dear referee

Enclosed you will find a copy of our revised manuscript, “Microorganisms- 3546331” entitled “Hygienic Quality of Air-Packed and Refrigerated or Frozen Stored Döner Kebab and Evaluation of the Growth of Listeria monocytogenes Intentionally Inoculated” submitted to Microorganisms.

The authors would like to thank the reviewers for their careful reading of the manuscript and the resulting constructive comments and suggestions. Basically, we agree with all the points raised by the reviewers, and wherever possible the manuscript has been modified as recommended. All reviewer comments are in black plain font, whereas our response is described in red plain font. We have made the changes and corrections based on the reviewer’s suggestions. We evaluated the comments and prepared a point-by-point response to each one of them.

Reviewer 2

The manuscript is a research article that delas with the shelf-life estimation of a traditional meat product, that is kebab, under different refrigerated conditions. Challenge tests were also carried out with pathogens. The manuscript has novel insights and falls within the aimsand scope of the journal. The authors state that numerous samples were analyzed which provides a strength for the study. The paper has been well prepared and data were treated with statistical analysis, one-way analysis of variance and Tukey's honestly significance test.

This should be better defined in the relevant section. My comments deal with the packaging used.

Why the authors used this multi-layer film?

Answer – Thanks, the multi-layer film has been chosen by the manufacturer, the same for the air packaging.

Did the authors run preliminary tests in other films?

Answer - Thanks

We did not run preliminary tests in other films because this was not the aim of the work. We thank you for this suggestion. Indeed this could be a future direction. The use of different plastic films and packaging such as modified atmosphere packaging, vacuum packaging and sous vide packaging could be investigated. However scientific literarature indicates that the growth – no growth of the microorganisms is influenced by the film but it is influenced by the packaging technology.

Another comment is that the authors should compare their data with studies using other packaging films. Pros and Cons in this context.

Answer – Thanks, I added some considerations

Lines 384-406 - Similar results were obtained by Bingöl et al. [3] in a work aimed at assessing the quality of air-packaged and vacuum-packaged frozen kebabs, using barrier foam trays (LLDPE/EVOH multilayer), wrapped with a polyethylene film for air packaging (AP) or heat-sealed with a Multivac packaging unit for vacuum packaging (VP) with multilayer polyolefin with PVDC film. Comparing our results to the above authors, this study indicates that the freezing process not only prevents microbial growth but also produces a reduction of microbial counts over time. According to Bingöl et al. [3], the reduction is higher in Vacuum Packaging kebabs than in air-packaged ones and TVC values of the product was significantly different (p < 0,05) between the product packaged in air and the VP ones. In addition, they showed that storage time and packaging conditions were the significant factors influencing TBARS values. No influence was attributed to the films. The TBARS values of all samples increased in up to 8 months of storage time, then decreased until the end of storage; levels in VP samples remained lower than those of AP ones during the entire storage time. It was observed that the evolution of microorganisms was not attributed to the film, but rather to the types of packaging utilised. The Sous Vide (SV) process was observed to ensure the microbiological safety of döner kebab products for a period of at least 100 days, while stored at 4 °C. The SV was the most effective technique among the investigated packaging methods on increasing the shelf-life during cold storage, without adding any preservatives or additives, and protected the microbial quality of döner kebabs 20, 6, and 5 times more than AP, MAP and VP, respectively [5].

Freezing, as expected, also inhibited the production of TVB-N and MDA in the investigated samples. The values observed at the end of the 30 days were largely acceptable (Table 3).

Finally, sensory anaysis of the treted product under refrigeration  is missing.

Answer - Thanks, we made it, but we wanted to use it in a future paper. We add it now.

Lines 232-247 - Lines Sensorial analyses were performed by 20 non-professional and non-trained assessors (10 women and 10 men, representing food technology students aged between 22 and 24 years of age). The choice of non-professional tasters was mandatory because they represent typical consumers. Sensory analysis was performed based on the triangle test (UNI EN ISO 4120:2004). Three lots of Döner kebab, each represented by 10 samples, were evaluated by tasters who were asked to evaluate the influence of the time of the refrigerated storage (0 days versus 30 days) on product quality. In short, three samples, two of which were identical, coded with three-digit numbers, were given in randomized service order, and the assessors were asked to find out the different ones. Each tested sample was cooked at 120 °C for 15 min in an oven. After cooking, the products were cooled at 65 °C and samples were presented, wrapped in aluminum foil, in a quiet room, and the answers were collected on a paper card

The assessors who identified the different samples were asked to indicate their preference. The scoring system used was (0 days versus 30 days): 1 (excellent), 2 (good), 3 (sufficient), and 4 (scarce). Statistical evaluation of the results was carried out according to Stone and Sidel [28]

Sensorial Analysis

Lines 513-518 - The twenty nonprofessional subjects were unable to distinguish the two types of Döner kebab (0 and 30 days storage). The triangle test methodology demonstrated that the refrigerated storage for 30 days did not influence the odor or flavor of the Döner kebab. So, they established that there was no difference between the two samples. This analysis confirmed the microbial and physico-chemical results and the acceptability of the product up to 30 days on refrigerated storage.

References added

Lines 611-613 - 27. ISO 4120:2004. Triangle Test Methodology. Standard Test Method for Sensory Analysis—General Guidance for the Design of test Rooms.

  1. Stone, H.; Sidel, J.L. Sensory Evaluation Practices, 3rd ed.; Elsevier Academic Press: San Diego, CA, USA, 2004

-
